# Oral aspirin for preventing colorectal adenoma recurrence: A systematic review and network meta-analysis of randomized controlled trials

Khanh Dinh Hoang[1,2], Jin-Hua Chen[3], Tsai-Wei Huang[4,5], Yi-No Kang[5,6,7], Chiehfeng Chen [5,8,9]*

1 International Master's Program in Medicine, College of Medicine, Taipei Medical University, Taipei, Taiwan, 2 Department of Histopathology, Hai Phong University of Medicine and Pharmacy, Hai Phong, Vietnam, 3 Graduate Institute of Data Science, Taipei Medical University, Taipei, Taiwan, 4 School of Nursing, Taipei Medical University, Taipei, Taiwan, 5 Cochrane Taiwan, Taipei Medical University, Taipei, Taiwan, 6 Research Center of the Big Data and Meta-Analysis Center, Wan Fang Hospital, Taipei Medical University, Taipei, Taiwan, 7 Institute of Health Policy & Management, College of Public Health, National Taiwan University, Taipei, Taiwan, 8 Department of Public Health, College of Medicine, Taipei Medical University, Taipei, Taiwan, 9 Division of Plastic Surgery, Department of Surgery, Wan Fang Hospital, Taipei Medical University, Taipei, Taiwan

* clifchen@tmu.edu.tw

**Data Availability Statement:** All relevant data are within the paper and its Supporting information files.

## Abstract

Colorectal adenomas have the potential of malignant transformation if left untreated. Multiple randomized controlled trials have been performed to evaluate the efficacy of aspirin in preventing colorectal adenoma recurrence in a population with a history of colorectal adenoma but not colorectal cancer, however, the relationship between aspirin dose and colorectal adenoma recurrence remains unclear. We conducted pairwise meta-analysis, meta-regression, trial sequential analysis, and network meta-analysis of all eligible studies. The ROB 2.0 tool was used to assess the risk of bias in the studies. The confidence in network meta-analysis (CINeMA) approach was used to evaluate the confidence of the network meta-analysis results. The network meta-analysis included eight RCTs (nine reports), comprising four on aspirin (low or high dose) alone and four on aspirin combined with another medication, all compared with placebo. In the network meta-analysis, low-dose aspirin (LDA <300 mg per day) was more effective than high-dose aspirin (HDA ≥300 mg per day) and placebo, with risk ratios of 0.76 (95% CI: 0.58 to 0.99) and 0.7 (95% CI: 0.54 to 0.91), respectively. LDA was the optimal treatment relative to HDA and placebo (P-score = 0.99). In the trial sequential analysis, LDA was only more effective than placebo when the number of included participants exceeded the optimal information size; this was not the case for HDA. LDA has statistically significant efficacy for colorectal adenoma prevention, but compared with HDA, its efficacy remains uncertain. Further trials are therefore required.

**Funding:** The authors received no specific funding for this work.

**Competing interests:** The authors have declared that no competing interests exist.

## Introduction

Colorectal adenomas are common precancerous neoplasms in the gastrointestinal tract, with an annual incidence rate of 7.2%, whereas the recurrence rate of nonadvanced and advanced adenomas was 19.3% and 22.9%, respectively according to a study in Japan in 2004 [1]. Colorectal 57 adenomas are polypoid lesions that develop in the colorectal lumen either sporadically (most cases) 58 or as a part of adenoma-related genetic syndromes [2]. If left untreated, these precursor lesions may 59 progress to invasive carcinoma through three main molecular mechanisms: chromosomal 60 instability, microsatellite instability, and CpG island methylator phenotype methylation [3]. In Europe in 2006, invasive carcinoma was second to breast cancer as the leading cause of death, with 412,900 cases (equivalent to an incidence rate of 12.9%) [4]. Generally, the progression rate from adenoma to invasive cancer depends on the histological subtype: 4.8%, 22.5%, and 40.7% for tubular adenoma, tubulovillous adenoma, and villous adenoma, respectively [5]. In addition, an adenoma transforms into invasive carcinoma between 4 and 48 years, depending on the polyp size and histologic morphology; adenomas or other polypoid lesions can be detected and removed easily through colonoscopy [6]. However, during colonoscopy, patients may experience unpleasant sensations (mainly varying degrees of pain); thus, approximately half the patients are unwilling to return for a surveillance procedure within the following 5 years [7].

Numerous studies have been conducted to develop methods to prevent colorectal adenomas without performing periodic colonoscopy, such as changing eating habits to include a high-fiber diet [8] or high dry-bean intake [9] or chemoprevention using multiple medications, including aspirin (acetylsalicylic acid) and other nonsteroidal anti-inflammatory drugs (e.g., celecoxib or rofecoxib) [10]. Some randomized controlled trials (RCTs) have indicated that low-dose aspirin (LDA, 81–160 mg per day) is more effective than high-dose aspirin (HDA, 300–325 mg per day) or selective cyclooxygenase enzyme (COX-2) inhibitors (e.g., 400 mg of celecoxib per day) [10]. However, no specific effective aspirin dose has been indicated in any clinical guidelines so far, except in one by Cancer Council Australia which recommends that aspirin dose of 100–300 mg per day can reduce the mortality rate in colorectal cancer but unrelated to colorectal adenomas [11]. Synchronously, most studies have not clarified the relationship between the aspirin dose and colorectal adenoma recurrence.

Therefore, the objective of this study is to clarify the relationship between the aspirin dose and colorectal adenoma recurrence and to determine the preferred aspirin dose range for clinical use for colorectal adenoma prevention.

## Methods

This systematic review and network meta-analysis was conducted in accordance with the Preferred Reporting Items for Systematic Reviews and Meta-Analyses 2015 checklist [12].

### Eligibility criteria

**Inclusion criteria.** We only included RCTs in which patients had a history of colorectal adenoma and had no contraindications for aspirin use, such as gastric bleeding or ulcer, hemophilia, or kidney dysfunction. Before each trial was initiated, participants had undergone colonoscopy to remove all observed polypoid lesions. Participants then received either aspirin or placebo on a daily basis, with a compliance of at least 70%. The adenomas included in the analysis were noninvasive lesions.

**Exclusion criteria.** We excluded all non-RCT studies (cohort, case–control, and cross-sectional studies). We also excluded RCTs with participants who did not receive aspirin or

placebo on a daily basis as well as those with evaluations of biomarkers or adenoma-related risk factors and any other nonadenoma recurrence-based studies.

## Search strategy and information sources

We searched the PubMed/MEDLINE, Scopus, and Embase databases for relevant studies every week (on Friday) to check for all possible updates. The last search was conducted on December 31, 2021. PubMed Medical Subject Headings terms and keywords were used, including "colon adenoma," "rectal adenoma," "colorectal adenoma," "colorectal polyp," "colorectal tumor," "aspirin," and "acetylsalicylic acid." No restriction on publication language or search time was applied to decrease the risks of reporting and publication bias.

## Study selection and data collection processes

Standardized checklists in which all the essential inclusion and exclusion criteria were listed in detail and clearly explained were used for eligible trial selection. Duplicated studies were removed using Endnote 20 [13]. Two reviewers (KDH and YNK) screened all the titles and abstracts independently, followed by full-text assessment of potentially eligible trials. The two reviewers also independently collected published data used for statistical analysis. Any discrepancies between the two reviewers (KDH and YNK) were resolved through a consensus or the adjudication of a third reviewer (CC).

## Data items and subgroup analysis

**Primary outcome.**   Overall colorectal adenoma recurrence was defined as the occurrence of any adenoma (any size or histologic subtype) in the colorectum after the removal of all observable lesions at the beginning of a trial at any aspirin dose and any time. Histologically, participants with low-grade or high-grade dysplasia adenoma or in-situ carcinoma were included in the analysis.

Adenoma recurrence was stratified according to aspirin dosage, namely HDA ($\geq$300 mg per day) and LDA ($<$300 mg per day); the intervention time of short-term aspirin use ($\leq$1 year) or long-term aspirin use ($>$1 year); and aspirin dose (six doses of 81, 100, 160, 300, 325, and 600 mg per day).

**Secondary outcome.**   The secondary outcome was the mean number of recurrent adenomas. This outcome was analyzed per patient, and all patients.

The incidence of severe adverse events was determined based on the number of participants who had at least one adverse event during follow-up.

## Study risk-of-bias assessment

The risk of bias of all the included studies was assessed using the ROB 2.0 tool [14]. The assessment results of each included study were saved in the supplementary file. The entire process was performed by two reviewers (DKH and YNK) independently. Disagreements were resolved through a consensus or based on the opinion of a third reviewer (CC). The ROB assessment results are displayed in a risk-of-bias summary figure (S1 Fig) created using Robvis [15]. Publication and reporting bias were assessed using a funnel plot [16].

## Effect measures and statistical analysis

**For pairwise meta-analysis.**   Revman 5.4 software [17] was used for the pairwise meta-analysis of all outcomes. To avoid overestimating the effect size and misunderstanding the results, we analyzed all dichotomous outcomes through a risk ratio estimated using the

Mantel–Haenszel method with a random-effects model instead of an odds ratio [18]. Continuous outcomes (mean number of recurrent adenomas) were analyzed based on the mean difference determined using the inverse variance method with a random-effects model. Between-study heterogeneity was assessed using $Tau^2$, $Chi^2$, and $I^2$ tests; 95% confidence interval (95% CI) and the P value of 0.05 were applied to all analyses (except $Chi^2$ with p value <0.10).

**For meta-regression.** We evaluated the relationship between the aspirin dose (individual doses) and dose response (the overall colorectal adenoma recurrence according to the risk ratio) using R 4.1.0 (Rstudio version 1.41717). Meta-regression was conducted using a random-effects model and both linear and nonlinear models. The model fit was also compared using analysis of variance.

**For network meta-analysis.** The confidence in network meta-analysis (CINeMA) approach was employed to assess the certainty of evidence in the network using a random-effects NMA model [19, 20]. We rated the certainty of evidence in each domain in each comparison as "no concerns," "some concerns," or "major concerns." To evaluate imprecision, heterogeneity, and incoherence, we defined the clinically key effect size (using the risk ratio) as 1.25 (the range of equivalence was defined from 0.8 to 1.25). Then, we judged the level of certainty across domains using the 4-level confidence of the GRADE approach: very low, low, moderate, or high [21].

"*A triangle of LDA, HDA, and placebo*," manifesting the interactive relationship between three interventions, was created using NMAstudio (version 0.1) [22], which is a web application programmed in Python and linked to the netmeta package in R software. Based on the results of the network meta-analysis, we ranked the treatments under study to determine the optimal treatment in the network by using the surface under the cumulative ranking curve measure (P-score). The results are presented in the form of a heatmap.

**For trial sequential analysis.** Additionally, we performed trial sequential analysis to make decision on whether or not to conduct further clinical trials [23]. Data pooling was performed based on the DerSimonian–Laird method, with boundaries based on the O'Brien–Fleming method, by using R software.

## Results

### Search strategy

By December 31, 2021, 7,918 articles from the three databases had been identified based on a screening of titles and abstracts. After the removal of duplicated articles and screening titles and abstracts, 50 articles were included in full-text assessment to determine their eligibility. Finally, 9 studies representing 8 RCTs [24–32] were included in the quantitative analysis (Fig 1). In 1993, Gann et al. [33] conducted a large clinical trial involving more than 22,000 participants (all the participants were male physicians). However, the participants did not receive aspirin or placebo on a daily basis and not all participants had undergone colonoscopy at the start of the trial. This study was therefore excluded.

### Trial characteristics

The eight trials (nine reports) that met the inclusion criteria were published between 2003 and 2021 and involved a total of 3,011 participants. Four of the trials (five reports) compared aspirin alone (LDA or HDA or both) with placebo, and the other four compared aspirin in combination with another medication (e.g., folic acid, mesalazine, resistant starch, or eicosapentaenoic acid) with double placebo. All the study participants were randomly assigned to the intervention arms, with 1,695 participants receiving aspirin and 1,316 receiving placebo. The mean age of participants with genetic syndromes (familial adenomatous polyposis or

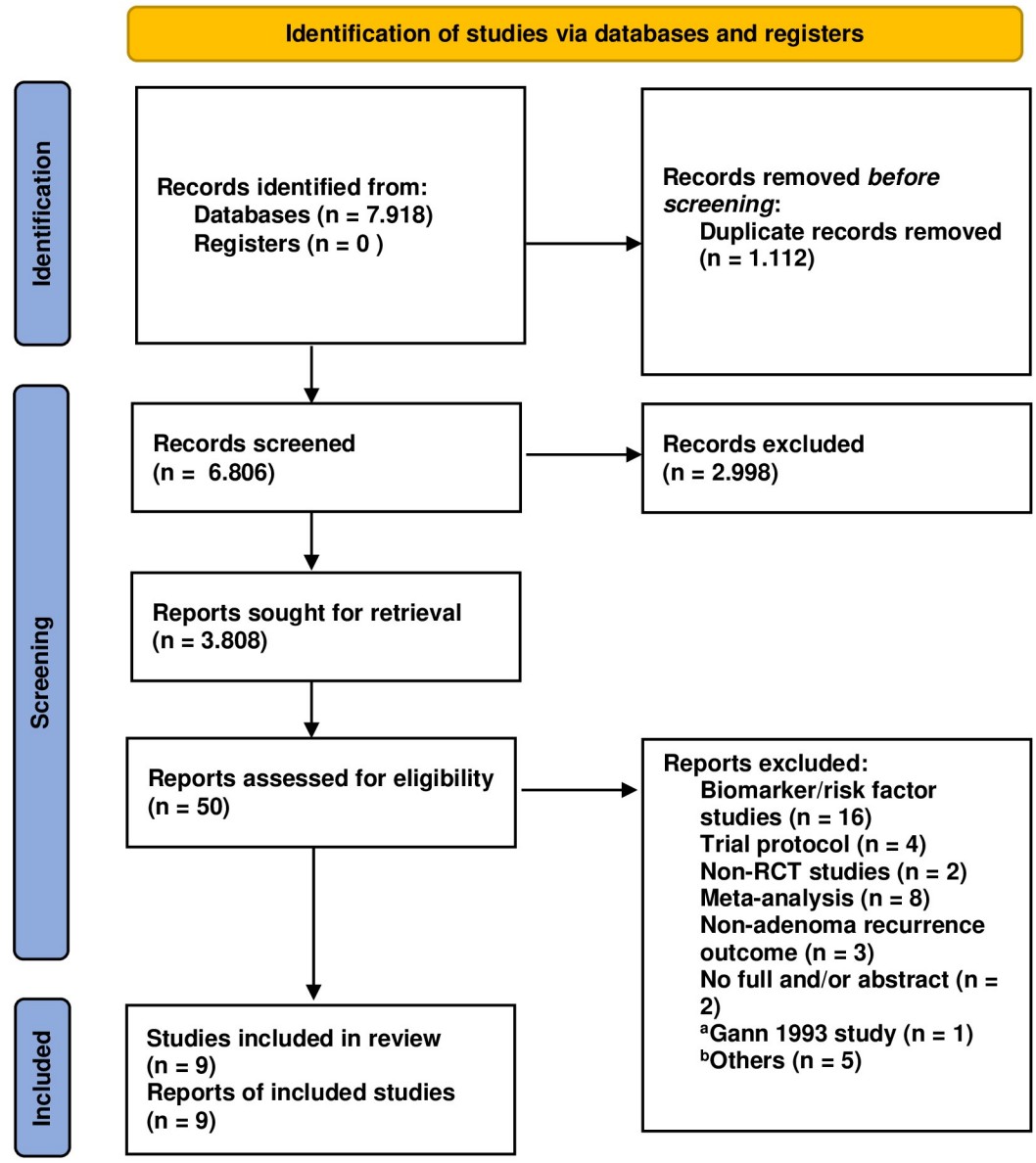

**Fig 1. Study search flow chart.** Note: [a]The reason to exclude this study was presented above. [b]The reasons for others were listed in Supporting information file.

Lynch syndrome) ranged from 30 to 59 years in three trials (Burn et al., [29] Ishikawa et al., [26] Ishikawa et al., [27] and the mean age of participants from the general population was 57–69 years. The follow-up time ranged from 8 months to 5 years. Seven trials (except the study by Ishikawa [26], which only provided data on severe adverse events) provided information on colorectal adenoma recurrence. Most trials examining the effects of different aspirin doses reported separate data for each dose, except for the study by Benamouzig et al. [31] (the data of two groups with aspirin doses of 160 and 300 mg were combined). In addition, one RCT [30, 31] was reported twice based on follow-up time (at year 1 and 4). However, for overall adenoma recurrence, we only included the study with the longest follow-up time. Characteristics of the trials included in this study were shown in Table 1.

**Table 1. Study characteristic.**

| Study name | Demographic features (mean age, gender and relevant genetic syndromes) | Study arm | Mean follow-up time | The outcomes of the trial |
|---|---|---|---|---|
| **Benamouzig 2003** [31], 2011 [30] | Gender (male: 190; female: 82). Age (aspirin: 57.8; placebo: 57.7 | Aspirin 300 mg<br>Aspirin 160 mg<br>Placebo | 1 year [31]<br>4 year [30] | The recurrence of any adenoma, advanced adenomas, mean number of recurrent adenomas, mean adenomatous polyp burden. |
| **Sandler 2003** [24] | Gender (male: 332; female: 303), (aspirin: 317; placebo: 318) | Aspirin 325 mg<br>Placebo | HDA: 30.9 months<br>Placebo: 31.6 months. | The recurrence of adenoma, total number of polyps in each group and the incidence of adverse events. |
| **Baron 2003** [32] | Gender (male: 712; female: 409). Age (LDA: 57.3; HDA: 57.7; placebo: 57.4) | Aspirin 325 mg<br>Aspirin 81 mg<br>Placebo | LDA: 32.5 months, HDA: 32.8 months, Placebo: 32.9 months | The recurrence of adenoma, the number of any adenoma, advanced lesions and the incidence of adverse events. |
| **Logan 2008** [25] | Gender (male: 534; female: 405). Age (Aspirin + folic acid: 56.9; aspirin: 57.9; folic acid: 58.3; placebo: 58). | Aspirin 300 mg + folic acid 0.5mg<br>Aspirin + folic acid placebo<br>Folic acid + Aspirin placebo<br>Double placebo | Aspirin + folic acid: 3.3 years, aspirin only: 3.4 years, folic acid only: 3.4 years, placebo: 3.4 years. | The recurrence of any adenoma, advanced adenoma, mean number of recurrent adenoma per patient and the incidence of adverse events. |
| **Burn 2008** [29] | Gender (male 336; female 410). Age (HDA + resistant starch: 44; HDA + placebo: 43; Resistant starch + placebo: 43; Placebo + placebo: 44), with Lynch syndrome. | Aspirin 600 mg + Resistant starch 30g<br>Aspirin + Resistant starch placebo<br>Resistant starch 30g + aspirin placebo<br>Double placebo | 29 months | The recurrence of adenoma, largest dimension of neoplasm, number of adenomas–number/total number for each study group and each agent. |
| **Ishikawa 2012** [26] | Gender (male: 17; female: 17). Age (LDA: 39.7; placebo: 36.7), with FAP syndrome. | Aspirin (100 mg)<br>Placebo | 10 months | The number of patients with reduced polyps, change in polyp height, mean diameter and number of polyps, and the incidence of adverse effects |
| **Hull 2018** [28] | Gender (male: 563; female: 144). Age (HDA + EPA: 66; HDA + EPA placebo: 65; HAD placebo + EPA: 65; HAD placebo + EPA placebo: 65). | EPA (2g) + aspirin (300 mg)<br>EPA + placebo<br>Aspirin + placebo<br>Double placebo | 1 year | The recurrence of any adenomas, advanced‡ adenomas, number of adenoma according to location, histologic subtype,. . .and the incidence of adverse events. |
| **Ishikawa 2021** [27] | Gender (male: 53; female: 49). Age (Aspirin + mesalazine: 31·5; Aspirin + placebo: 38; Mesalazine + placebo: 35·5; Placebo + placebo: 30), with FAP syndrome. | Aspirin (100mg) + mesalazin (2g)<br>Aspirin + placebo<br>Mesalazine + placebo<br>Double placebo | 8 months | The recurrence of adenoma, mean number of adernoma, number of adenoma, maximum polyp size and number of polyps (any size) in each part of colon. The incidence of adverse events. |

Note that: LDA: Low-dose aspirin, HDA: High-dose aspirin; PLB: Placebo; FAP syndrome: Familial adenomatous polyposis syndrome, EPA: Eicosapentaenoic acid.

## Primary outcome and subgroup analysis

**Risk-of-bias assessment (for the primary outcome).** Of seven included RCTs, four had an overall low risk of bias [26–29, 32], one had some concerns [25] related to missing outcome data (more than 10% of participants withdrew from the trial mainly because of death, poor health status, or lack of contact with the participants), and two had an overall high risk of bias because they deviated from the intended interventions and had missing outcome data (use of inappropriate analysis and too many eligible participants were excluded from the analysis [24]) and differences between intervention groups in the proportions of missing outcome data (28.45% in the placebo group and 20.31% in the aspirin group [30, 31]). The risk of bias in the

included studies is presented in S1 Fig. The funnel plot for the primary outcome appeared relatively symmetrical (S2 Fig), indicating no evidence of publication bias.

**Overall adenoma recurrence (primary outcome).** Eight reports (seven RCTs) provided data on overall adenoma recurrence [24, 25, 27–32]. In general, aspirin significantly reduced the overall recurrence of colorectal adenoma at any dose and any time during follow-up (risk ratio [RR] = 0.86, 95% CI: 0.75 to 0.99, P = 0.03, and $I^2$ = 40%) (Fig 2). A subgroup analysis revealed no significant difference between the aspirin and placebo groups for short-term aspirin use (RR = 0.81, 95% CI: 0.59 to 1.10, P = 0.18, and $I^2$ = 64%); however, for long-term aspirin use, a statistically significant difference was detected between the two groups (RR = 0.84, 95% CI: 0.73 to 0.98, P = 0.02, and $I^2$ = 24%).

**Adenoma recurrence with low-dose aspirin use.** Only three RCTs provided data on adenoma recurrence with LDA use [27, 30, 32]. Data from two reports (Benamouzig 2003 et al., [31] and Benamouzig 2011 et al., [30]) were based on the same trial, however, Benamouzig 2003 combined the two groups of 160 mg aspirin and 300 mg aspirin into a single aspirin group. While, Benamouzig 2011 reported into two groups "Analysis at 1 or 4 years" and "Analysis at 4 years", which led to the inability to collect data for short-term aspirin use. At any time,

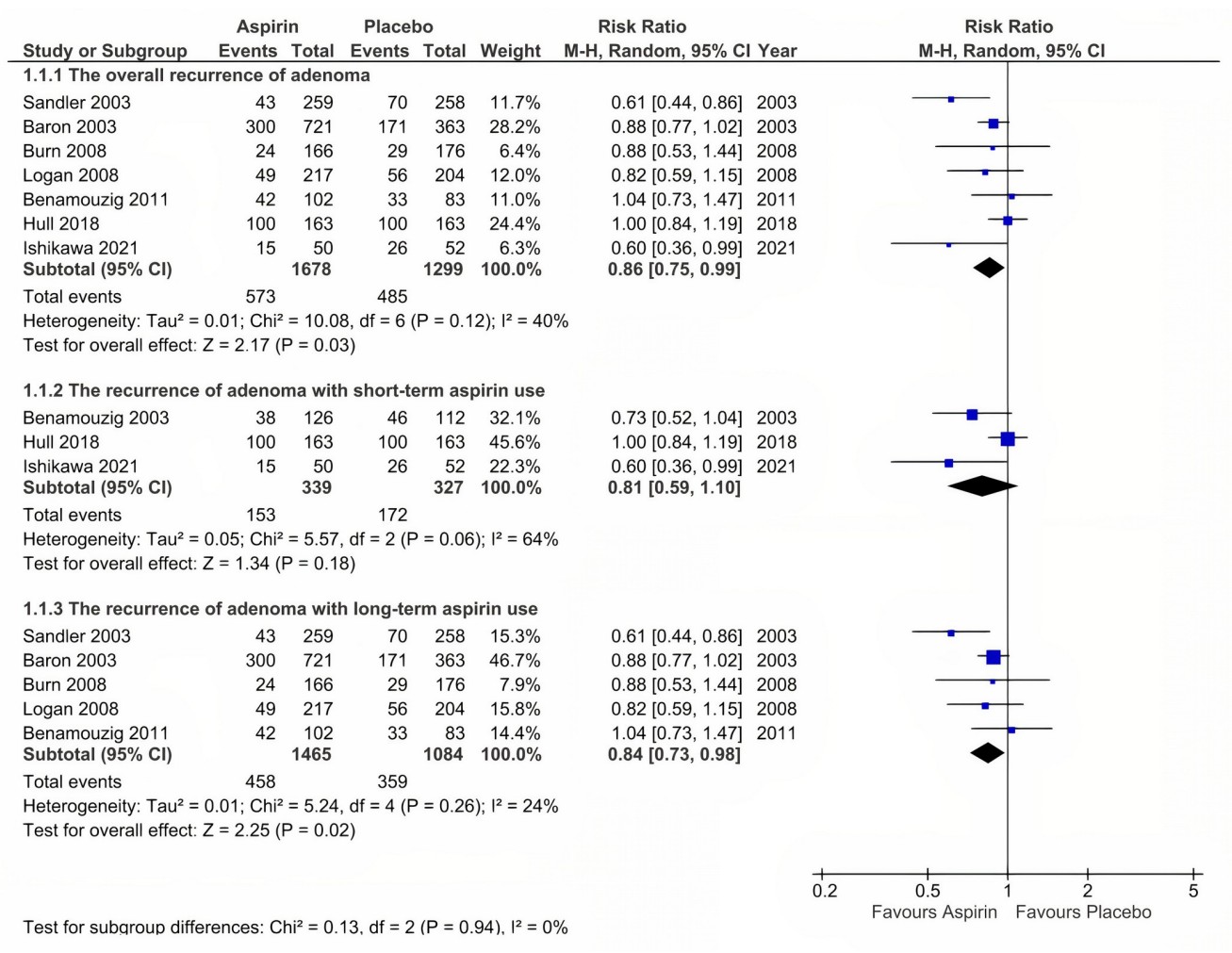

**Fig 2. Overall adenoma recurence (the primary outcome).**

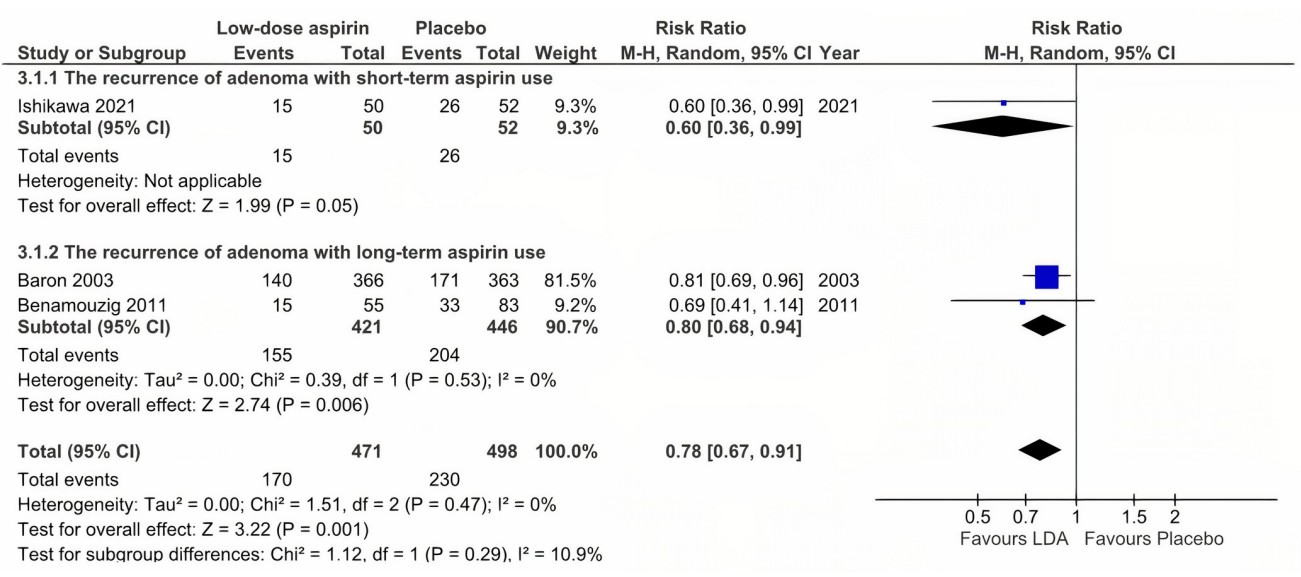

**Fig 3. Adenoma recurrence with low-dose aspirin use.**

LDA decreased adenoma recurrence by 22% (RR = 0.78, 95% CI: 0.67 to 0.91, P = 0.001, and $I^2$ = 0%) (Fig 3). Moreover, the subgroup analysis revealed the same results for both short-term and long-term use of LDA (RR = 0.60, 95% CI: 0.36 to 0.99, and P = 0.05; $I^2$ was not applicable because only one RCT was included vs. RR = 0.80, 95% CI: 0.68 to 0.94, P = 0.006, and $I^2$ = 0%).

**Adenoma recurrence with high-dose aspirin use.** Six RCTs were included in the analysis of the effect of HDA use [24, 25, 28–30, 32]. No difference was observed between the aspirin and placebo groups when participants were treated with HDA (RR = 0.93, 95% CI: 0.77 to 1.11, P = 0.43, and $I^2$ = 62%) (Fig 4). In addition, the subgroup analysis revealed no significant difference between the aspirin and placebo groups in terms of either short-term or long-term use (RR = 1.00, 95% CI: 0.84 to 1.19, and P = 1.00; $I^2$ was not applicable because only one trial was included vs. RR = 0.91, 95% CI: 0.71 to 1.16, P = 0.44, and $I^2$ = 68%).

**Comparison of low-dose and high-dose aspirin use in relation to adenoma recurrence.** For this comparison, two RCTs were included in the analysis [30, 32]. Both RCTs provided data on colorectal adenoma prevention with long-term aspirin use. One of these two trials had a high risk of bias resulting from an inappropriate analysis in which too many eligible participants were excluded from the analysis. The analysis indicated that there were no statistically significant difference between (RR = 0.67, 95% CI: 0.38 to 1.17, P = 0.16, and $I^2$ = 79%) (Fig 5).

**Sensitivity analysis for the primary outcome.** After two studies with a high risk of bias (Sandler et al. [24] and Benamouzig et al. [30]) were removed from the overall adenoma recurrence outcome analysis, the results indicated that aspirin became less effective over time (RR = 0.9, 95% CI: 0.80 to 1.00, P = 0.05, and $I^2$ = 9% vs. RR = 0.86, 95% CI: 0.75 to 0.99, and P = 0.03 for the original analysis), but this was nonsignificant. We also performed a similar analysis of the effect of LDA on colorectal adenoma prevention and identified no significant difference between aspirin and placebo groups and an increased inconsistency between studies (RR = 0.77, 95% CI: 0.61 to 0.97, P = 0.02, and $I^2$ = 20% vs. RR = 0.78, 95% CI: 0.67 to 0.91, P = 0.001, and $I^2$ = 0%).

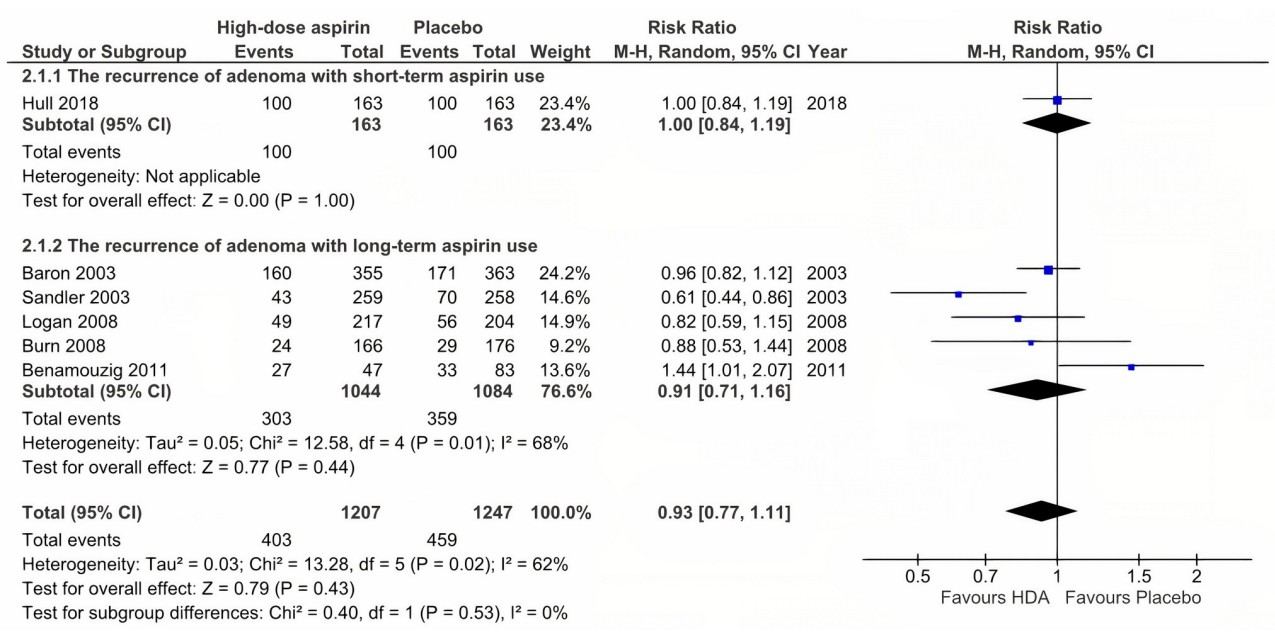

**Fig 4. Adenoma recurrence with high-dose aspirin use.**

## Secondary outcome

**Mean number of recurrent adenomas.** Data from six reports representing five RCTs were pooled for this outcome analysis [24, 25, 27, 28, 30, 31]. The analysis favored aspirin over placebo (mean difference −0.16, 95% CI: −0.26 to −0.06, P = 0.001, and $I^2$ = 0%) (S3 Fig).

**Incidence of severe adverse events.** Data from six RCTs were pooled for this analysis [24, 26, 28–30, 32]. The analysis revealed that more severe adverse events were observed in the placebo group than in the aspirin group (RR = 1.29, 95% CI: 1.05 to 1.59, P = 0.02, and $I^2$ = 0%), indicating that aspirin may be an unsafe drug for the prevention of adenoma recurrence, especially LDA (S4 Fig).

**Sensitivity analysis for incidence of severe adverse events.** For severe adverse events, of six RCTs [24, 26, 28–30, 32]. included in the analysis, the study by Baron et al. [32] reported a significantly higher rate of severe adverse events in both the aspirin and placebo groups than other studies (data shown in S4 Fig). After this study was excluded from the analysis, no statistically significant difference in the incidence of severe adverse events in either the aspirin or placebo group was detected (RR = 0.99, 95% CI: 0.69, 1.43), indicating that aspirin might be a safe drug for use at both low and high doses for preventing colorectal adenomas.

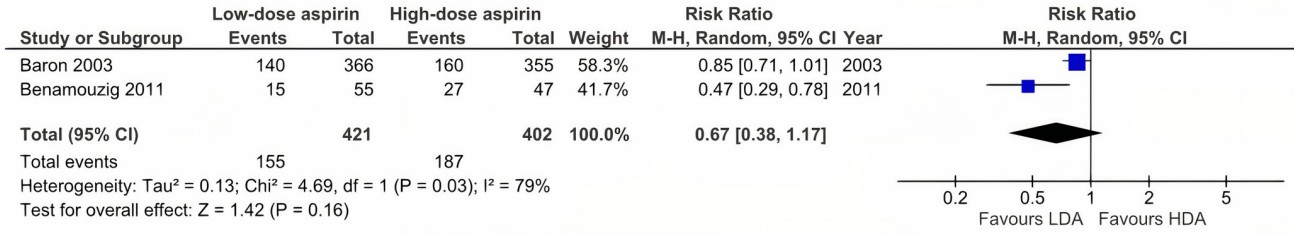

**Fig 5. Comparison of low-dose and high-dose aspirin use in relation to adenoma recurrence.**

## Meta-regression

Data for the meta-regression analysis were extracted from seven RCTs [24, 25, 27–30, 32]. According to the linear line (red dotted line), the most effective aspirin dose fluctuated by approximately 100 mg (RR 0.6, 95% CI: 0.36 to 0.99), and a downward trend in the effectiveness of aspirin with the increasing aspirin dose was noted. Furthermore, an analogous predisposition was observed in the left branch of the blue parabola. However, the relationship was not statistically significant in either linear or nonlinear models ($P > 0.05$) (S5 Fig).

## Network meta-analysis to compare low-dose and high-dose aspirin use and placebo

Data from seven RCTs were pooled for this analysis [24, 25, 27–30, 32]. The analysis revealed that LDA was favored over HDA and PLB with statistcally significant difference (NMA estimate: RR = 1.32, 95% CI: 1.01 to 1.73, and RR = 0.70, 95% CI: 0.54 to 0.91, respectively shown in League table (S6 Fig)), with moderate certainty of evidence for LDA vs PLB comparison due to some concerns in heterogeineity, but with low certainty of evidence for LDA vs HDA comparison due to major concerns in heterogeneity (shown in S7 Fig); finally, no substantial difference were observed between HDA and PLB (RR = 0.92, 95% CI: 0.77 to 1.10) with low level of certainty due to within-study bias, imprecision, and heterogeneity. The network plot of low-dose, high-dose aspirin and placebo was shown in Fig 6.

In the P-score heatmap, the treatments (LDA, HDA, and placebo) were classified by colour from most effective to least effective (from yellow to violet), with a P-score ranging from 0 to 1, which can be interpreted as an average degree of certainty for a treatment to be more effective than other treatments in the network. The heatmap suggests that compared with HDA and placebo, LDA is the most effective treatment in the network with P-score = 0.99, whereas P-score = 0.42 for HDA and P-score = 0.1 for PLB (Fig 7).

## Trial sequential analysis

For the LDA versus placebo comparison, the last point of the Z-curve lying outside the monitoring boundary favored LDA, and the number of participants included in the analysis exceeded the required information size, indicating that the difference between the two groups was significant, and that no further trials were required (Fig 8). By contrast, no significant difference between the HDA and placebo groups or between the HDA and LDA groups was recorded in this analysis (S14 and S15 Figs).

## Discussion

Overall, our findings indicated that LDA was beneficial in preventing colorectal adenoma recurrence and that long-term usage of LDA was more preventive than short-term use.

From the pairwise analyses, we encountered that aspirin usage (at any dose compared with placebo) lowered the risk of colorectal adenoma recurrence by roughly 14% and that long-term aspirin use resulted in a statistically significant difference from short-term use. In addition, aspirin also assisted in reducing the number of recurrent adenomas that developed after the intervention, and this effect was more significant with long-term use of aspirin than with short-term use.

Network meta-analyses proved that LDA was more efficient than HDA in terms of effectiveness. Additionally, when compared to HDA and placebo, LDA was assessed as the most successful treatment in the network (as shown in the heatmap in Fig 8). Therefore, we could infer that LDA was superior to HDA for the long-term prevention of colorectal adenoma.

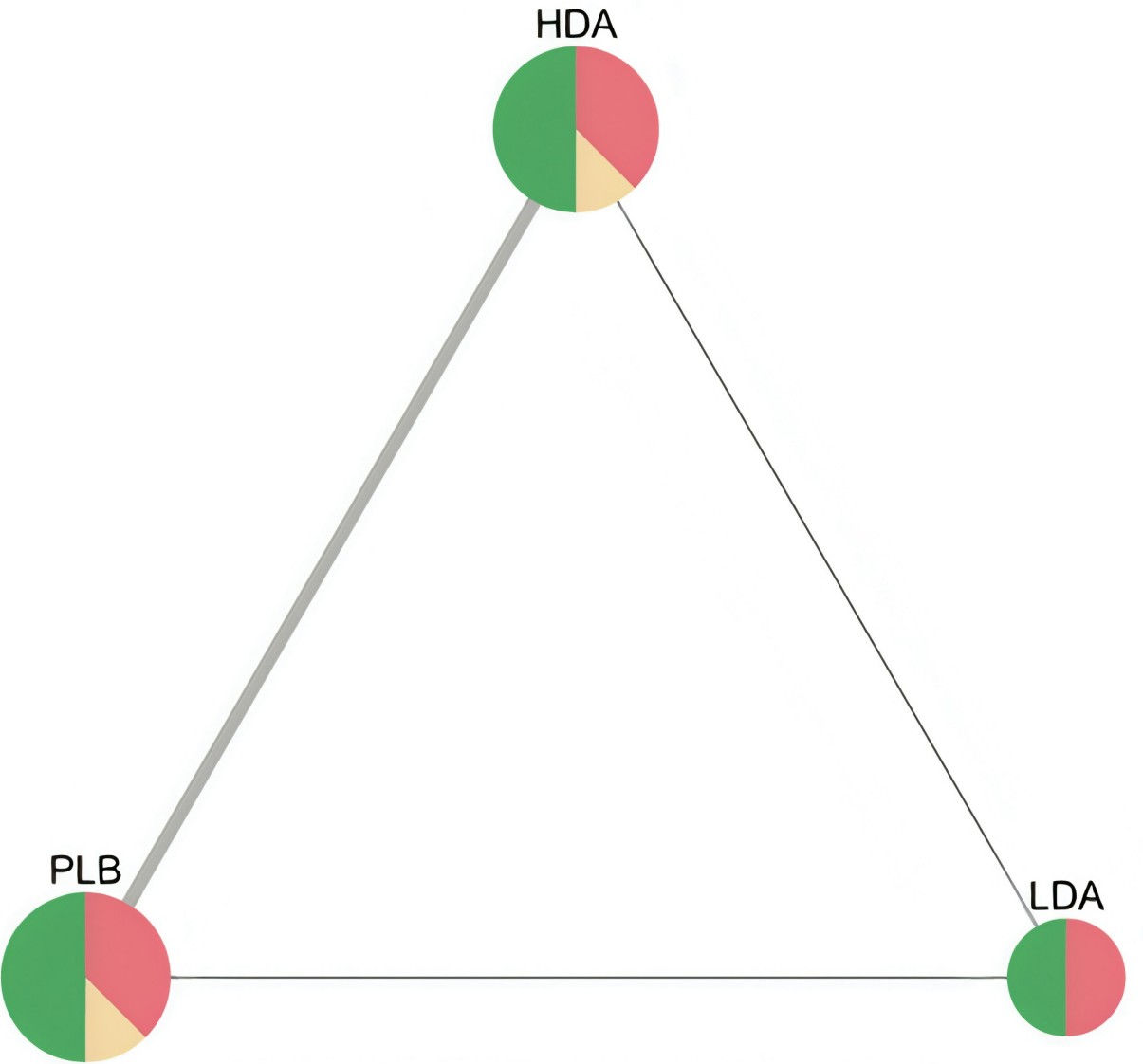

**Fig 6. Network plot of low-dose, high-dose aspirin and placebo in RCTs for recurrence of colorectal adenomas.** Note that node size is proportional to sample size, edge width is proportional to number of studies, node colour refers to majority ROB. (HDA: high-dose aspirin, LDA: low-dose aspirin, PLB: placebo).

Veettil et al., [10] confirmed the role of LDA in colorectal adenoma recurrence, however, they merely compared LDA and HDA indirectly via the mediating effect of placebo.

The optimal information size for the analysis was attained, according to the existing trial sequential analysis, and LDA was found to be more effective than placebo (Fig 6). However, no statistically significant difference was found between HDA and placebo, or between LDA and HDA (S14 and S15 Figs). This appeared to be at odds with the network meta-analysis's findings. Furthermore, the number of participants in the comparison of LDA and HDA surpassed the required information size, while the sample size in the comparison of HDA and placebo remained modest.

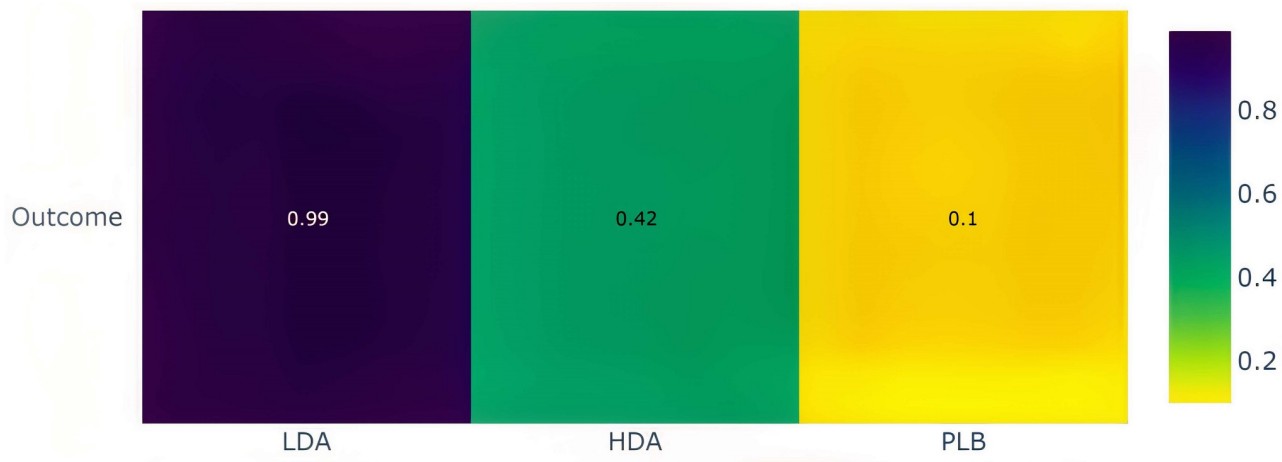

**Fig 7. The heatmap of treatment ranking of the netwwork.** Note that LDA: low-dose aspirin, HDA: high-dose aspirin, PLB: placebo. P-score ranges from 0 to 1 (equivalent to color ranging from yellow to violet).

In the meta-regression, we found that the effectiveness of aspirin diminished with increasing dose, signifying that LDA (particularly LDA of about 100 mg) might be the most promising treatment. Although we were unable to fully explain this tendency, we presume that this finding is valuable for clinical practice. One recent study recommended using aspirin at a dose of less than 100 mg to avoid advanced lesions [34]. Furthermore, within a secure dose range, a lower dose generally offers more advantages over higher doses, including reduced costs (economic effectiveness), fewer side effects, and greater therapeutic efficacy [35]. The lack of a threshold of effectiveness for clinical usage, however, might make it challenging to determine a particular dose in practice (a too low dose might not have any benefits).

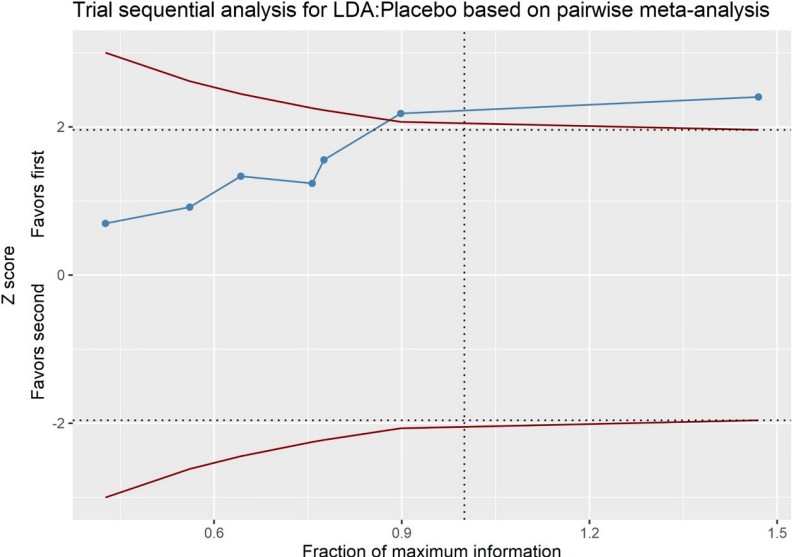

**Fig 8. Trial sequential analysis (TSA).** Note that blue line, red lines, horizontal dotted lines, vertical dotted lines represent Z-curve, monitoring boundary, required information size, conventional boundary, respectively. The areas outside the red lines indicate the area favoring LDA or placebo. LDA: Low-dose aspirin.

According to the results of the aforementioned analyses, LDA was, especially when used over a long period of time, more efficient than placebo in colorectal adenoma prevention. Besides, LDA was verified relatively effective in preventing cardiovascular disorders [36]. We believe that LDA (less than 300 mg per day) should be applied clinically to prevent the recurrence of numerous pathologic conditions, including colorectal adenomas. Based on trial sequence analysisa and meta-regression, it appears that there was inadequate evidence to determine whether LDA or HDA was more protective in the prevention of colorectal adenoma recurrence. Hence, more clinical studies must be conducted to assess LDA's effectiveness in comparison to HDA. However, given the high number of participants in our analysis, this might not be possible, which would necessitate more trials to produce a substantial shift.

Baron et al., [32], which was a large trial with a rigid study design, identified a very high rate of severe adverse events in the aspirin group compared to those in other studies (some favored aspirin and some favored placebo but with a nonsignificant difference), indicating that aspirin is a detrimental drug (RR = 1.29, 95% CI: 1.05 to 1.59). While the follow-up period was relatively lengthy (almost three years), the authors included all hospitalizations as major adverse events, leading to a significant number of serious adverse effects without providing specific justifications. It was therefore unable to affirm with certainty that those events were induced by aspirin or other illnesses picked up during follow-up. The discrepancy in serious adverse events between the two groups became negligible once this study was excluded from the analysis, suggesting that aspirin may be a safe drug for clinical use (RR = 0.99, 95% CI: 0.69, 1.43).

Our analysis still has certain limitations despite the inclusion of a few new trials and the fresh strategy for steadfast statistical analyses. The credibility of the findings could be impacted by the relatively small number of included studies and the lack of repeated direct comparisons between HDA and LDA (only two RCTs were provided). Additionally, some trials had relatively small sample sizes and inconsistent intervention periods. Furthermore, the notable risk of bias in several of the included studies might have lowered the dependability of the findings. Eventually, this analysis could only include RCTs assessing aspirin effectiveness for up to 5 years rather than for a longer period.

## Conclusion

Compared with placebo, LDA significantly reduces the recurrence of colorectal adenomas in participants with and without adenoma-related syndromes. However, its efficacy compared with HDA remains uncertain. Further RCTs are therefore required to verify these results.

## Supporting information

**S1 Fig. Risk of bias assessments for the primary outcome.**
(TIF)

**S2 Fig. The funnel plot for assessing publication bias in the main outcome.**
(TIF)

**S3 Fig. The mean number of recurrent adenomas.**
(TIF)

**S4 Fig. The number of participants experiencing serious adverse events.**
(TIF)

**S5 Fig. The relationship between aspirin dose and colorectal adenoma recurrence.**
(TIF)

**S6 Fig. League table.**
(TIF)

**S7 Fig. Certainty of evidence from Network meta-analysis.**
(TIF)

**S8 Fig. Risk of bias contributions.**
(TIF)

**S9 Fig. Risk of bias distribution in the included interventions in the network.**
(TIF)

**S10 Fig. Indirectness in the network.**
(TIF)

**S11 Fig. Imprecision assessments.**
(TIF)

**S12 Fig. Heterogeneity assessments.**
(TIF)

**S13 Fig. Incoherence assessments.**
(TIF)

**S14 Fig. Trial sequential analysis (TSA).**
(TIF)

**S15 Fig. Trial sequential analysis (TSA).**
(TIF)

**S1 Checklist. PRISMA NMA checklist of items to include when reporting a systematic review involving a network meta-analysis.**
(DOCX)

## Acknowledgments

This manuscript was edited by Wallace Academic Editing.

## Author Contributions

**Conceptualization:** Khanh Dinh Hoang, Yi-No Kang, Chiehfeng Chen.

**Data curation:** Khanh Dinh Hoang, Yi-No Kang, Chiehfeng Chen.

**Formal analysis:** Khanh Dinh Hoang, Jin-Hua Chen, Tsai-Wei Huang, Yi-No Kang, Chiehfeng Chen.

**Funding acquisition:** Chiehfeng Chen.

**Investigation:** Khanh Dinh Hoang, Yi-No Kang, Chiehfeng Chen.

**Methodology:** Khanh Dinh Hoang, Jin-Hua Chen, Tsai-Wei Huang, Yi-No Kang, Chiehfeng Chen.

**Project administration:** Chiehfeng Chen.

**Resources:** Khanh Dinh Hoang.

**Software:** Khanh Dinh Hoang, Yi-No Kang, Chiehfeng Chen.

**Supervision:** Chiehfeng Chen.

**Validation:** Khanh Dinh Hoang, Yi-No Kang, Chiehfeng Chen.

**Visualization:** Khanh Dinh Hoang, Yi-No Kang, Chiehfeng Chen.

**Writing – original draft:** Khanh Dinh Hoang, Jin-Hua Chen, Tsai-Wei Huang, Yi-No Kang, Chiehfeng Chen.

**Writing – review & editing:** Khanh Dinh Hoang, Jin-Hua Chen, Tsai-Wei Huang, Yi-No Kang, Chiehfeng Chen.

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
