## [Decision Letter · Decision Letter 0]

8 Mar 2023

PONE-D-22-33838Oral aspirin for preventing colorectal adenoma recurrence: a systematic review and network meta-analysis of randomized controlled trialsPLOS ONE

Dear Dr. Chen,

Thank you for submitting your manuscript to PLOS ONE. After careful consideration, we feel that it has merit but does not fully meet PLOS ONE’s publication criteria as it currently stands. Therefore, we invite you to submit a revised version of the manuscript that addresses the points raised during the review process.

We look forward to receiving your revised manuscript.

Kind regards,

Tariq Jamal Siddiqi

Academic Editor

PLOS ONE

Journal Requirements:

Reviewers' comments:

Reviewer's Responses to Questions

**Comments to the Author**

1. Is the manuscript technically sound, and do the data support the conclusions?

Reviewer #1: Yes

2. Has the statistical analysis been performed appropriately and rigorously? 

Reviewer #1: Yes

3. Have the authors made all data underlying the findings in their manuscript fully available?

Reviewer #1: Yes

4. Is the manuscript presented in an intelligible fashion and written in standard English?

Reviewer #1: Yes

5. Review Comments to the Author

Reviewer #1: The authors have conducted a comprehensive systematic review and network meta-analysis of RCTs to evaluate the prophylactic role of oral aspirin for colorectal adenoma recurrence. While commendable effort has been done for the extensive methodology outlined and explained in the article, and figures and tables are appropriately designed, there are some major points that the authors must consider for improvement.

1. The introduction of the article is unnecessarily long and filled with excessive jargon which is not really required. For instance, the details regarding the pathophysiology is not needed in the introduction and can be removed. The authors should cut down on the text of the introduction and make it as comprehensive and readable as possible. Consider the following sequence for a better introduction - first mention the known details with relevant statistics and what the current guidelines state about oral aspirin for colorectal adenoma recurrence, then mention the conflicts in literature and what is the basis for investigating this and finally develop strongly on the rationale of the meta-analysis and mention the main objective of the study.

2. Consider including a subheading for conclusion and mention the conclusive statements there.

3. The discussion does not elaborate upon the future directions for oral aspirin use for preventing colorectal adenoma recurrences. Kindly add a paragraph for it.

4. The authors state in the first paragraph of discussion, 'its efficacy was more statistically significant for long-term use than for short-term use.' Please rephrase this sentence. Just saying more statistically significant does not indicate any direction of the results.

5. The authors state 'These results are not easy to explain' in the discussion while mentioning the large trial by Baron et al. This is a very important study and given the fact that it is well-designed and robustly conducted, removing this study from the analysis should have a solid reason which must be stated in the text. Please also discuss why the risk of severe adverse events was significantly higher in the aspirin group in this study. While it may be hard to state definitive reasons, the authors can provide suggestive reasons as to why this was the case in this large trial.

6. The discussion lacks details about clinical implications of this meta-analysis. Kindly add a paragraph for it.

6. PLOS authors have the option to publish the peer review history of their article (what does this mean?). If published, this will include your full peer review and any attached files.

Reviewer #1: No

---

## [Author Response · Author response to Decision Letter 0]

18 Apr 2023

I wrote a "Response to Reviewers" and uploaded to the system.

---

## [Editor Report · Decision Letter 1]

6 Sep 2023

PONE-D-22-33838R1Oral aspirin for preventing colorectal adenoma recurrence: a systematic review and network meta-analysis of randomized controlled trialsPLOS ONE

Dear Dr. Chen,

Thank you for submitting your manuscript to PLOS ONE. After careful consideration, we feel that it has merit but does not fully meet PLOS ONE’s publication criteria as it currently stands. Therefore, we invite you to submit a revised version of the manuscript that addresses the points raised during the review process.

We look forward to receiving your revised manuscript.

Kind regards,

Rodney John Scott

Academic Editor

PLOS ONE

Journal Requirements:

Additional Editor Comments:

The manuscript needs to be corrected for English language and poor grammatical construction. Once a revised manuscript is submitted that does not include grammatical errors, I will send it out for re-review.

---

## [Author Response · Author response to Decision Letter 1]

10 Sep 2023

Dear Reviewers, 

First of all, we would like to thank you for your consideration and your valuable comments on our manuscript. Thanks to your comments, we realize that our manuscript still has some shortcomings. We have tried to perfect our manuscript as best we could based on your comments.

Comment #1: Please review your reference list to ensure that it is complete and correct. If you have cited papers that have been retracted, please include the rationale for doing so in the manuscript text, or remove these references and replace them with relevant current references. Any changes to the reference list should be mentioned in the rebuttal letter that accompanies your revised manuscript. If you need to cite a retracted article, indicate the article’s retracted status in the References list and also include a citation and full reference for the retraction notice. 

Authors: We thoroughly reviewed the reference lists multiple times according to your comment and found that they are totally accurate. Up to now, none of these references have been retracted from online database. Therefore, we kept the reference list remain unchanged.

Additional Editor Comments: The manuscript needs to be corrected for English language and poor grammatical construction. Once a revised manuscript is submitted that does not include grammatical errors, I will send it out for re-review.

Authors: We found several problems with poor grammartical structures and English language, so we rewrote some paragraphs in the 'Discussion' section. Any modifications were highlighted in 'Revised Manuscript with Track Changes'.

One more time, thank you for your consideration of this manuscript. We hope our manuscript to receive good responses from you and then to be accepted by the journal.

Sincerely,

Chiehfeng Chen MD. MPH. PhD.

---

## [Editor Report · Decision Letter 2]

13 Sep 2023

Oral aspirin for preventing colorectal adenoma recurrence: a systematic review and network meta-analysis of randomized controlled trials

PONE-D-22-33838R2

Dear Dr. Chen,

We’re pleased to inform you that your manuscript has been judged scientifically suitable for publication and will be formally accepted for publication once it meets all outstanding technical requirements.

Kind regards,

Rodney John Scott

Academic Editor

PLOS ONE
---

## [Editor Report · Acceptance letter]

18 Sep 2023

PONE-D-22-33838R2 

Oral aspirin for preventing colorectal adenoma recurrence: a systematic review and network meta-analysis of randomized controlled trials 

Dear Dr. Chen:

I'm pleased to inform you that your manuscript has been deemed suitable for publication in PLOS ONE. Congratulations! Your manuscript is now with our production department. 

Kind regards, 

on behalf of

Dr. Rodney John Scott 

Academic Editor

PLOS ONE